# Pre-Learning Stress That Is Temporally Removed from Acquisition Impairs Fear Learning

**DOI:** 10.3390/biology12060775

**Published:** 2023-05-26

**Authors:** Phillip R. Zoladz, Chloe N. Cordes, Jordan N. Weiser, Kassidy E. Reneau, Kayla M. Boaz, Sara J. Helwig, Emma M. Virden, Caitlin K. Thebeault, Cassidy L. Pfister, Bruktawit A. Getnet, Taylor D. Niese, Sydney L. Parker, Mercedes L. Stanek, Kristen E. Long, Seth D. Norrholm, Boyd R. Rorabaugh

**Affiliations:** 1Psychology Program, The School of Health and Behavioral Sciences, Ohio Northern University, Ada, OH 45810, USA; c-cordes@onu.edu (C.N.C.); j-weiser.2@onu.edu (J.N.W.); k-reneau.2@onu.edu (K.E.R.); k-boaz@onu.edu (K.M.B.); s-helwig@onu.edu (S.J.H.); e-virden@onu.edu (E.M.V.); c-thebeault@onu.edu (C.K.T.); c-pfister.1@onu.edu (C.L.P.); b-getnet@onu.edu (B.A.G.); t-niese.1@onu.edu (T.D.N.); s-parker@onu.edu (S.L.P.); m-stanek.1@onu.edu (M.L.S.); k-long.7@onu.edu (K.E.L.); 2Department of Psychiatry and Behavioral Neurosciences, School of Medicine, Wayne State University, Detroit, MI 48202, USA; snorrholm@wayne.edu; 3Department of Pharmaceutical Sciences, School of Pharmacy, Marshall University, Huntington, WV 25755, USA; rorabaughb@marshall.edu

**Keywords:** stress, cortisol, fear conditioning, startle, generalization

## Abstract

**Simple Summary:**

Understanding how stress impacts fear learning can provide important insight into the etiology of stress-related psychological disorders, such as post-traumatic stress disorder (PTSD). Thus, we exposed healthy participants to a brief stressor 30 min prior to a learning task in which one visual stimulus (CS+), but not another (CS−), was associated with an aversive airblast to the throat. The next day, we quantified participants’ fear in response to the CS+, CS−, and several visual stimuli that had never been observed by participants. Our results indicated that stress impaired the acquisition of fear on Day 1, particularly in participants who exhibited the greatest cortisol responses to stress. Contrary to expectations, pre-learning stress did not significantly influence the generalization of fear measured on Day 2. Our findings suggest that stress, perhaps through increased cortisol levels, alters the acquisition of fear. This may be useful in understanding the distortion of fear memories in stress-related disorders.

**Abstract:**

Few studies have examined the time-dependent effects of stress on fear learning. Previously, we found that stress immediately before fear conditioning enhanced fear learning. Here, we aimed to extend these findings by assessing the effects of stress 30 min prior to fear conditioning on fear learning and fear generalization. Two hundred and twenty-one healthy adults underwent stress (socially evaluated cold pressor test) or a control manipulation 30 min before completing differential fear conditioning in a fear-potentiated startle paradigm. One visual stimulus (CS+), but not another (CS−), was associated with an aversive airblast to the throat (US) during acquisition. The next day, participants were tested for their fear responses to the CS+, CS−, and several generalization stimuli. Stress impaired the acquisition of fear on Day 1 but had no significant impact on fear generalization. The stress-induced impairment of fear learning was particularly evident in participants who exhibited a robust cortisol response to the stressor. These findings are consistent with the notion that stress administered 30 min before learning impairs memory formation via corticosteroid-related mechanisms and may help us understand how fear memories are altered in stress-related psychological disorders.

## 1. Introduction

The effects of pre-learning stress on long-term memory are particularly relevant for our understanding of eyewitness and traumatic memories. However, these effects have been inconsistent in the research literature, with studies reporting enhancements, impairments, or no effects on memory. More recently, research has shown that the impact of pre-learning stress on long-term memory depends largely on the temporal proximity of the stressor to the learning experience, although other factors, such as the sex of the participants and the type of task being learned, play a role as well [1,2,3,4,5]. Studies in this area suggest that when stress is administered shortly before learning, long-term memory is enhanced (e.g., [1,6,7,8,9,10,11,12]), while stress that is temporally separated from learning impairs long-term memory (e.g., [6,10,13,14]). Research from our laboratory and from others has suggested that the time-dependent effects of pre-learning stress on long-term memory are associated with an amygdala-mediated biphasic effect of corticosteroids on synaptic plasticity in cognitive brain regions, such as the amygdala and hippocampus [1,2,15,16,17]. According to this view, shortly after stress, corticosteroids exert rapid, non-genomic effects that, in conjunction with a rapid increase in norepinephrine, are excitatory in nature and enhance amygdala and hippocampus activity [1,6,7,8,9,10,18,19]. However, as the stress response continues, corticosteroids exert delayed, inhibitory effects on these brain areas, resulting in impaired learning and memory [1,10,13,19]. Neuroimaging work has supported this view by showing that IV administration of hydrocortisone rapidly (within 5 min) enhances hippocampus and amygdala activity while reducing the activity 20–25 min later [20].

Most of the work examining the time-dependent effects of pre-learning stress on long-term memory has involved neutral, non-arousing material, such as word lists or images. Few studies have investigated the impact of pre-learning stress on emotional learning, such as fear conditioning, a type of learning that is particularly appropriate for our understanding of stress-related psychological disorders, such as post-traumatic stress disorder (PTSD). The limited clinical literature that does exist regarding stress effects on fear conditioning is conflicting, with studies reporting enhancements or impairments of fear learning [21,22,23,24,25,26,27]. Furthermore, only a small number of studies of which we are aware have manipulated the timing of stress before fear learning [22,28]. Antov and colleagues showed that stress immediately before fear conditioning increased extinction resistance, conceivably by producing a stronger fear memory [22]. In contrast, when stress was administered 30–40 min prior to fear conditioning, participants’ cortisol responses negatively correlated with their fear responses, suggesting a cortisol-mediated impairment of fear memory. Previous work from our own laboratory produced results that were consistent with Antov et al.’s [22] first manipulation and showed that immediate, pre-learning stress enhanced differential conditioning in a fear-potentiated startle paradigm [12]. Nevertheless, not all investigators have reported similar findings [28].

Research specifically addressing the impact of stress that is temporally separated from learning on fear memory has also been inconsistent. Merz et al. [21] found that stress administered 40 min before fear conditioning impaired fear learning in males while having no effect in females. Similar to the findings observed by Antov and colleagues [22], stress-induced cortisol levels in males were negatively associated with the strength of their fear learning [21]. Other studies have reported that administering stress long before fear conditioning enhances or has no effect on fear learning [24,28]. Despite contradictory findings in this area of research, the results of Antov et al. [22] and Merz et al. [21] are consistent with the proposed mechanisms for the time-dependent effects of pre-learning stress on long-term memory outlined above, namely, that when stress is administered 30 min or more before learning, it exerts impairing effects on cognitive processes via the delayed increase in corticosteroid levels exerting a negative impact on synaptic plasticity in cognitive brain regions. Given the inconsistent findings, however, additional work that tests this hypothesis is warranted.

Studies investigating the impact of stress on fear conditioning may aid our understanding of stress-related psychological disorders. Indeed, extensive work has revealed abnormal fear conditioning processes in people with an array of psychological disorders, most notably PTSD [29,30]. However, researchers have proposed that overgeneralization of fear, an exaggerated form of the classical conditioning phenomenon stimulus generalization, is a robust, more selective pathogenic marker of clinical anxiety [31,32,33,34]. Stimulus generalization involves an organism exhibiting a conditioned response to stimuli that are perceptually or semantically similar to a conditioned stimulus. Generalization of fear is adaptive because it allows an organism to generalize fear to novel stimuli that may present a threat to survival. However, broad generalization, and thus an inability to distinguish between threat and safety, can be burdensome to daily life. Research has demonstrated an overgeneralization of fear in many psychological disorders that involve pathological anxiety (e.g., PTSD, panic disorder, generalized anxiety disorder, social anxiety disorder) [31,34,35,36,37]. Thus, in the present study, we examined the impact of stress administered 30 min before fear conditioning on fear acquisition and fear generalization.

There is a limited collection of studies that have examined the effects of stress on fear generalization. For the most part, these studies suggest that stress enhances the generalization of fear. In one preclinical study, stress or the infusion of corticosterone directly into the hippocampus after fear conditioning resulted in increased fear to an incorrect predictor of threat [38]. In humans, two investigations have reported that stress administered prior to generalization testing enhanced fear generalization [39,40] (however, see [41] for conflicting findings). Only a couple of studies have investigated the impact of pre-learning stress on fear generalization. One preclinical study reported that stress exposure 24 h prior to fear learning led to greater fear generalization, but in humans, Sep et al. [28] found that stress administered 2 h before conditioning had no impact on fear generalization. Although findings have been mixed, the stress-induced enhancement of fear generalization has been attributed to its impact on pattern separation [33,42], a function that is heavily reliant on the hippocampus [43]. In the present study, we hypothesized that if stress is administered 30 min prior to conditioning, it would impair fear acquisition (and pattern separation abilities) due to its time-dependent inhibitory impact on amygdala and hippocampus function. This would result in a weaker memory and, thus, greater fear generalization to non-threatening stimuli. If such a stress-induced impairment of fear learning is dependent on the delayed increase in corticosteroid levels, these effects should also be associated with salivary cortisol in participants.

## 2. Materials and Methods

### 2.1. Participants

The data presented in this manuscript represent a subset of data from a larger study examining the influences of acute stress, sex, and childhood maltreatment on fear learning and fear generalization. The data presented here are independent of other published manuscripts [44] based on the larger dataset. Two hundred and twenty-one healthy undergraduate students (81 males, 140 females; age: *M* = 19.23, *SD* = 1.89) from Ohio Northern University volunteered to participate in the experiment. Individuals were excluded from participating if they met any of the following conditions: diagnosis of Raynaud’s or peripheral vascular disease; diagnosis of post-traumatic stress disorder (PTSD); presence of skin diseases, such as psoriasis, eczema, or scleroderma; history of syncope or vasovagal response to stress; history of any heart condition or cardiovascular issues (e.g., high blood pressure); history of severe head trauma; current treatment with narcotics, beta-blockers, or steroids; pregnancy; substance use disorder; regular use of recreational drugs; regular nightshift work; auditory disorder; hearing impairment. Participants were asked to refrain from drinking alcohol or exercising extensively for 24 h prior to the experimental sessions and to refrain from eating or drinking anything but water for 2 h prior to the experimental sessions. All experimental procedures were approved by the Institutional Review Board at Ohio Northern University, carried out in accordance with the Declaration of Helsinki, and undertaken with the understanding and written consent of each participant. Participants were awarded class credit and 20 USD upon completion of the study. 

### 2.2. Experimental Procedures

All experimental procedures took place between 1000 and 1700 h.

#### 2.2.1. Socially Evaluated Cold Pressor Test (SECPT)

After completing a short demographics survey and providing baseline physiological responses (see below), participants placed their dominant hand in a bath of water for 3 min. Participants randomly assigned to the stress condition (*N* = 93; 37 males, 56 females) placed their hand in ice cold (0–2 °C) water; participants randomly assigned to the control condition (*N* = 128; 44 males, 84 females) placed their hand in lukewarm (35–37 °C) water. The temperature of the water was maintained by a circulating water bath (Cole-Parmer; Vernon Hills, IL, USA). Participants were asked to keep their hand in the water for 3 min, but if it was too painful, the participant was allowed to remove his or her hand and continue with the experiment. Twenty-two participants from the stress condition removed their hand from the water before 3 min had elapsed (*M* = 67.00 s, *SD* = 39.65); importantly, these participants did not differ from those participants who kept their hand in for the entire 3 min on any dependent measure. All participants from the control condition kept their hand in the water for 3 min. Participants in the stress condition were also misleadingly told that they were being videotaped during the procedure to allow researchers to subsequently evaluate their facial expressions. These participants were asked to keep their eyes on a camera (located on the wall of the laboratory) throughout the manipulation. Prior work has shown that the SECPT results in increased heart rate, blood pressure, and salivary cortisol levels [8,10,11,12,13,19,45]. Moreover, the SECPT results in a greater physiological stress response than the standard cold pressor test, which lacks a social evaluative component.

#### 2.2.2. Subjective and Objective Stress Response Measures

*Subjective pain and stress ratings*. Participants rated the painfulness and stressfulness of the water SECPT at 1 min intervals on 11-point scales ranging from 0 to 10, with 0 indicating a complete lack of pain or stress and 10 indicating unbearable pain or stress.

*Cardiovascular analysis*. Heart rate (HR) was measured for 1 min before the SECPT until its completion. A BioNomadix pulse transducer (Biopac Systems, Inc.; Goleta, CA, USA) was placed on the ring finger of participants’ non-dominant hand [12]. The transducer was connected to the PPG module of the MP150 Biopac hardware. Average baseline HR (average of 1 min before SECPT) and water bath HR (average of SECPT) were calculated for statistical analysis.

*Salivary cortisol analysis*. On Day 1, saliva samples were collected from participants immediately before and 25 min after the water bath to analyze salivary cortisol concentrations [8,9,11,12,13,18,19,46]. On Day 2, saliva samples were collected from participants immediately before and after fear generalization (see below) to analyze salivary cortisol levels. Saliva samples were collected in a SalivaBio Oral Swab (Salimetrics, LLC, State College, PA, USA) that was placed under participants’ tongues for approximately 1.5 min. The samples were stored at −20 °C until being thawed and extracted via low-speed centrifugation. Salivary cortisol levels were then determined by an investigator who was blind to the conditions of participants via enzyme immunoassay (EIA; Cayman Chemical Co., Ann Arbor, MI, USA) according to the manufacturer’s protocol.

#### 2.2.3. Differential Fear Conditioning Paradigm

The differential fear conditioning paradigm that was used in the present study followed that which has been studied extensively in previous work [47,48,49,50,51,52,53,54] but with modified stimuli and a modified timeline. Unlike much of the previous work with this paradigm, each fear-potentiated startle session was separated by a period of 24 h, as in our own previously published work [12]. The paradigm included fear-potentiated startle [via electromyographic (EMG) recordings] and electrodermal activity (EDA) as the primary dependent measures and consisted of two phases: Day 1—fear acquisition and Day 2—fear generalization.

*Stimuli*. The startle probe was a 40 ms, 108 dB white noise burst that was delivered through headphones. The conditioned stimuli (CSs) were two black circles (filled with white) presented on the white background of a computer monitor (SuperLab software; Cedrus Corporation, San Pedro, CA, USA) that was positioned in front of participants. For half of the participants, the CS+ was a small circle (8 cm diameter), and the CS− was a large circle (28 cm diameter); for the other half of participants, the CS+ was the large circle, and the CS− was the small circle. The unconditioned stimulus (US) was a 250 ms, 140 psi blast of air that was directed at the throat. This US has been used in several studies and reliably produces fear-potentiated startle (e.g., [12,47,48,49,50,51,52,53,54]). On CS+ trials, the startle probe occurred 6 s after the onset of the CS, which was followed 500 ms later by presentation of the US; the CS+ terminated 500 ms following the onset of the US. On CS− trials, the startle probe occurred 6 s after the onset of the CS, without any presentation of the US; the CS− terminated 250 ms following onset of the startle probe. On noise-alone (NA) trials, the startle probe was presented alone as participants stared at the computer monitor; NA trials were the length of the startle probe (i.e., 40 ms).

*Psychophysiological measurement*. We measured the eyeblink component of the startle response by obtaining EMG recordings of the right orbicularis oculi muscle. Ag/AgCl electrodes were placed 1 cm below the pupil of the right eye, 1 cm under the lateral canthus, and behind the right ear over the mastoid bone (ground). For each participant, impedance levels were less than 6 kΩ. We measured EDA by placing two Ag/AgCl electrodes on the hypothenar surface of the participant’s non-dominant hand. The EMG and EDA data were obtained by using Acqknowledge data acquisition and analysis software (Biopac Systems, Inc.). The data were sampled at 1 kHz and amplified via the EMG and EDA modules of the Biopac MP150 system. 

*US expectancy measurement*. A response keypad (SuperLab software, Cedrus Corporation) was used to collect trial-by-trial ratings of US expectancies during each fear-potentiated startle session. During each CS presentation, participants pressed one of three buttons: an “AIR” key when they expected an airblast, a “NO AIR” key when they did not expect an airblast, and a “?” key when they were unsure what to expect. For data analysis, the responses of “AIR” were scored as +1, responses of “?” were scored as 0, and responses of “NO AIR” were scored as −1 [12,47,48,49,50,51,52,53,54].

*Days 1–2 (fear acquisition, fear generalization)*. Thirty minutes after the SECPT (Section 2.2.1), participants completed fear acquisition. We chose this time point because previous work from our laboratory has shown that acute stress administered 30 min prior to learning impairs long-term memory [10,13,19]. The acquisition phase began with three NA trials, followed by a habituation segment that consisted of four CS+, four CS−, and four NA trials. During this segment, none of the CSs were reinforced with an airblast US. After the habituation segment, participants underwent the conditioning phase. This phase consisted of three blocks of trials, with each block including four trials of each stimulus type (CS+, CS−, NA), resulting in 12 trials per block and 36 total trials. During conditioning, the CS+ was always followed by the airblast. 

On Day 2 of the experiment, participants underwent fear generalization testing. This phase began with three NA trials. Then, participants were exposed to 3 blocks of 10 different trials [1 for each of the following stimuli: CS+, 7 different generalization stimuli (GSs), CS−, NA] for a total of 30 trials. The 7 different GSs formed a size gradient (each differing by 2.5 cm diameter) between the small and large circles that were used as the CS+ and CS− during acquisition, similar to previous work (e.g., [35,43,55,56]). None of the CS presentations during the generalization phase were reinforced with an airblast US.

The same trial order was used for all participants. For acquisition, there had to be 4 trials of each trial type (i.e., CS+, CS−, NA) during each block of 12 trials. For generalization testing, there had to be 1 trial of each trial type (i.e., CS+, 7 GSs, CS−, NA) during each block of 10 trials. To prepare the trial order, trial type sequences were randomized within each trial block. The intertrial intervals were also randomized between 9 and 22 s. Figure 1 shows the timeline, stimuli, and trial block composition that made up each session.

*Psychophysiological data preprocessing*. MindWare EMG and EDA analysis programs (MindWare Technologies, Ltd., Gahanna, OH, USA) were used to filter and rectify the EMG and EDA signals that were obtained from the Acqknowledge data acquisition and analysis software. The EMG signal was amplified with a gain of 2000 and filtered with low- and high-frequency cutoffs at 28 and 500 Hz, respectively. We also applied a 60 Hz notch filter. The resultant data were exported for statistical analysis. We used the peak EMG signal 20–200 ms after each startle probe as a measure of participants’ acoustic startle responses [53,57,58,59,60]. The EDA measurements were used to compute participants’ skin conductance response (SCR). The SCR was defined as the maximum change in EDA (relative to a 1 s pre-CS baseline average) between 3 and 6 s after onset of the CS [47,61,62,63]. The SCRs were square-root-transformed before statistical analysis.

### 2.3. Statistical Analyses

To determine whether participants’ cortisol responses to the stress influenced its effects on fear learning and fear generalization, we divided stressed participants into “Responders” and “Non-Responders” based on their cortisol responses to the SECPT. Participants exhibiting a cortisol increase of at least 1.5 nmol/l following the SECPT were considered responders (*N* = 27; 13 males, 14 females); all other stressed participants were considered non-responders (*N* = 66; 24 males, 42 females). The cutoff for dividing participants into responder and non-responder groups was based on previous work from our laboratory [10,13] and from that of other investigators [64] who used a similar criterion.

We used mixed-model ANOVAs to analyze the subjective pain/stress ratings of the SECPT, HR, and cortisol concentrations, with condition (responder, non-responder, no stress) and sex (male, female) serving as the between-subjects factors and time point of measurement serving as the within-subjects factor. Similar to previous work (e.g., [12,48,49,50,52,65]), we used a difference score [(startle magnitude to the CS+ or CS− in each block)—(startle magnitude to the NA trials in each block)] to quantify fear-potentiated startle. This enabled us to calculate fear-potentiated startle relative to each participant’s baseline startle response (i.e., NA trials) and is supported by previous work showing its superiority to standardized difference scores and percent change scores [66]. Because startle responses can vary significantly within an individual, we calculated difference scores for each trial type within each block during acquisition (i.e., we averaged startle responses from the 4 trials of each trial type) and across all 3 blocks during generalization (i.e., we averaged startle responses from the 3 trials of each trial type) to obtain a more representative measure of each participants’ fear-potentiated startle [12,67,68,69]. For ease of comparison, we used a similar method for participant SCRs and US expectancies.

**Table 1 biology-12-00775-t001:** Pain and stress ratings of the SECPT.

Title 1	Minute 1	Minute 2	Minute 3
**Painfulness (scale of 0–10)**			
* Responders *			
Males	6.27 (0.41)	5.62 (0.46)	5.46 (0.50)
Females	5.79 (0.40)	6.21 (0.45)	6.50 (0.48)
* Non-Responders *			
Males	6.67 (0.30)	6.71 (0.34)	6.83 (0.37)
Females	6.48 (0.23)	6.88 (0.26)	7.10 (0.29)
* No Stress *			
Males	0.21 (0.22)	0.11 (0.25)	0.16 (0.27)
Females	0.16 (0.16)	0.11 (0.18)	0.13 (0.20)
**Stressfulness (scale of 0–10)**			
* Responders *			
Males	6.15 (0.52)	5.15 (0.57)	5.08 (0.60)
Females	4.79 (0.50)	5.43 (0.55)	5.50 (0.58)
* Non-Responders *			
Males	6.25 (0.38)	6.33 (0.42)	6.04 (0.44)
Females	5.98 (0.29)	6.12 (0.31)	6.19 (0.33)
* No Stress *			
Males	0.34 (0.28)	0.32 (0.31)	0.30 (0.33)
Females	0.42 (0.20)	0.41 (0.22)	0.39 (0.24)

Data are presented as means ± SEM.

We used separate mixed-model ANOVAs to analyze baseline startle responses (i.e., responses to the first 3 NA trials); fear-potentiated startle during habituation, conditioning, and generalization; SCR during habituation, conditioning, and generalization; and US expectancies during habituation, conditioning, and generalization. In these analyses, condition and sex served as the between-subjects factors, and, for the analyses of fear-potentiated startle, SCR, and US expectancies, stimulus (CS+, CS−) and trial block (only during acquisition) served as the within-subjects factors.

Alpha was set at 0.05 for all analyses, and Bonferroni-corrected post hoc tests were employed when the omnibus *F* indicated the presence of a significant effect. If the assumption of sphericity was violated, Greenhouse–Geisser corrections were employed, with reduced degrees of freedom reported in the analyses.

## 3. Results

### 3.1. Subjective and Objective Stress Response Measures

#### 3.1.1. Subjective Pain and Stress Ratings

Stressed participants, independent of cortisol response to the SECPT, rated the water bath as significantly more painful (effect of condition: *F*(2,215) = 421.24, *p* < 0.001, η^2^_p_ = 0.80) and stressful (effect of condition: *F*(2,215) = 212.57, *p* < 0.001, η^2^_p_ = 0.66) than non-stressed participants (see Table 1). The pain ratings in female non-responders increased throughout the SECPT (min 1 v. min 2: *p* = 0.018; min 1 v. min 3: *p* = 0.003), while the pain ratings in male responders decreased throughout the SECPT (min 1 v. min 2: *p* = 0.041; min 1 v. min 3: *p* = 0.052) (Sex × Time Point interaction: *F*(1.41,302.31) = 9.28, *p* < 0.001, η^2^_p_ = 0.04; Condition × Sex × Time Point interaction: *F*(2.81,302.31) = 3.53, *p* < 0.05, η^2^_p_ = 0.03). The stress ratings in male responders decreased throughout the SECPT (min 1 v. min 2: *p* = 0.006; min 1 v. min 3: *p* = 0.025) (Sex × Time Point interaction: *F*(1.42,305.66) = 8.08, *p* < 0.01, η^2^_p_ = 0.04; Condition × Sex × Time Point interaction: *F*(2.84,305.66) = 4.12, *p* < 0.01, η^2^_p_ = 0.04). No other main effects or interactions were significant (all *F* < 2.24, all *p* > 0.06).

#### 3.1.2. Cortisol

Because salivary cortisol levels were used to divide participants into responders and non-responders, the levels were first analyzed with stress serving as the between-subjects factor. This analysis revealed that stressed participants exhibited a significant increase (*p* < 0.001) in salivary cortisol levels following the SECPT, resulting in levels that were greater than non-stressed participants (*p* = 0.007) (effect of time point: *F*(1,205) = 12.44, *p* < 0.001, η^2^_p_ = 0.06; Stress × Time Point interaction: *F*(1,205) = 8.89, *p* < 0.01, η^2^_p_ = 0.04; see Figure 2). No other main effects or interactions were significant (all *F* < 3.09, all *p* > 0.08). The analysis using responder (i.e., condition) as a between-subjects factor unsurprisingly revealed that responders displayed a significant increase (*p* < 0.001) in salivary cortisol levels following the SECPT, resulting in levels that were greater than non-responders (*p* = 0.004) and non-stressed participants (*p* < 0.001) (effect of time point: *F*(1,203) = 52.87, *p* < 0.001, η^2^_p_ = 0.21; Condition × Time Point interaction: *F*(2,203) = 36.11, *p* < 0.001, η^2^_p_ = 0.26). No other main effects or interactions were significant (all *F* < 1.77, all *p* > 0.17). No significant main effects or interactions were observed when Day 2 salivary cortisol levels were analyzed with stress as the between-subjects factor (all *F* < 1.67, all *p* > 0.19). When Day 2 salivary cortisol levels were analyzed with responder as the between-subjects factor, there was a significant Condition × Sex × Time Point interaction, *F*(2,205) = 4.93, *p* < 0.01, η^2^_p_ = 0.05, suggesting that male responders exhibited a significant decrease in salivary cortisol levels after generalization testing (*p* = 0.014). No other main effects or interactions were significant (all *F* < 0.88, all *p* > 0.41).

#### 3.1.3. Heart Rate

To demonstrate that stress, overall, led to greater autonomic arousal, HR was first analyzed with stress serving as the between-subjects factor. This analysis showed that stressed participants exhibited a significance increase (*p* < 0.001) in HR during the SECPT, resulting in HR that was significantly greater than the HR observed in non-stressed participants (*p* = 0.025) (effect of time point: *F*(1,212) = 15.02, *p* < 0.001, η^2^_p_ = 0.07; Stress × Time Point interaction: *F*(1,212) = 23.40, *p* < 0.001, η^2^_p_ = 0.10; see Figure 2). Females also exhibited significantly greater HR, overall, than males (effect of sex: *F*(1,212) = 3.96, *p* = 0.048, η^2^_p_ = 0.02). No other main effects or interactions were significant (all *F* < 1.96, all *p* > 0.16). The analysis using responder (i.e., condition) as a between-subjects factor showed that both responders (*p* < 0.001) and non-responders (*p* < 0.001) exhibited significant increases in HR during the SECPT, while non-stressed participants did not (*p* > 0.47) (effect of time point: *F*(1,210) = 23.53, *p* < 0.001, η^2^_p_ = 0.10; Condition × Time Point interaction, *F*(2,210) = 11.80, *p* < 0.001, η^2^_p_ = 0.10). No other main effects or interactions were significant (all *F* < 1.66, all *p* > 0.19).

### 3.2. Fear Acquisition

#### 3.2.1. Fear-Potentiated Startle

Non-responders exhibited significantly greater baseline startle responses than responders (*p* = 0.015) and non-stressed participants (*p* = 0.014) (effect of condition: *F*(2,194) = 5.62, *p* < 0.01, η^2^_p_ = 0.06; see Figure 3). No other main effects or interactions were significant (all *F* < 0.58, all *p* > 0.44).

Analysis of fear-potentiated startle responses during the habituation phase indicated that non-responders displayed significantly greater overall (i.e., across both the CS+ and CS−) fear-potentiated startle responses than non-stressed participants (*p* < 0.001) (effect of condition: *F*(2,192) = 7.00, *p* < 0.001, η^2^_p_ = 0.07). No other main effects or interactions were significant (all *F* < 1.12, all *p* > 0.32).

Analysis of fear-potentiated startle responses during the conditioning phase revealed that participants exhibited significantly greater fear-potentiated startle responses to the CS+ than to the CS− during blocks 2 (*p* < 0.001) and 3 (*p* < 0.001) of conditioning (effect of stimulus: *F*(1,192) = 32.03, *p* < 0.001, η^2^_p_ = 0.14; Stimulus *×* Trial interaction: *F*(2,384) = 25.80, *p* < 0.001, η^2^_p_ = 0.12). This effect was dependent on condition. Across all three blocks of conditioning trials, both non-responders (*p* < 0.001) and non-stressed participants (*p* < 0.001) exhibited significantly greater fear-potentiated startle responses to the CS+ than to the CS−, but responders (*p* > 0.33) did not (Condition *×* Stimulus interaction: *F*(2,192) = 3.05, *p* = 0.05, η^2^_p_ = 0.03). No other main effects or interactions were significant (all *F* < 2.07, all *p* > 0.12).

#### 3.2.2. Skin Conductance Response

Analysis of SCRs during the habituation phase revealed no significant main effects or interactions (all *F* < 2.84, all *p* > 0.06).

Analyses of SCRs during the conditioning phase were similar to those observed for fear-potentiated startle. Across all three blocks of conditioning, participants exhibited significantly greater SCRs to the CS+ than to the CS− (effect of stimulus: *F*(1,191) = 5.50, *p* < 0.05, η^2^_p_ = 0.03; effect of trial: *F*(1.89,360.45) = 27.82, *p* < 0.001, η^2^_p_ = 0.13; see Figure 3). This effect depended on the condition (effect of condition: *F*(2,191) = 3.76, *p* < 0.05, η^2^_p_ = 0.04; Condition × Stimulus interaction: *F*(2,191) = 3.47, *p* < 0.05, η^2^_p_ = 0.04). Across all three blocks of conditioning trials, only non-stressed participants (*p* < 0.001) displayed significantly greater SCRs to the CS+ than to the CS− (responders: *p* > 0.78; non-responders: *p* > 0.14). Responders (*p* = 0.016), but not non-responders (*p* = 0.074), demonstrated significantly lower SCRs to the CS+ than did non-stressed participants. Females also exhibited significantly greater SCRs, overall, than males (effect of sex: *F*(1,191) = 3.92, *p* < 0.05, η^2^_p_ = 0.02). No other main effects or interactions were significant (all *F* < 2.24, all *p* > 0.10).

#### 3.2.3. US Expectancy Ratings

During the habituation phase, US expectancy ratings for the CS+ were significantly greater than US expectancy ratings for the CS−, despite the lack of any US reinforcement during this block of trials (effect of stimulus: *F*(1,202) = 5.34, *p* < 0.05, η^2^_p_ = 0.03; see Figure 4). No other main effects or interactions were significant (all *F* < 2.49, all *p* > 0.11).

During the conditioning phase, US expectancy ratings for the CS+ were significantly greater than US expectancy ratings for the CS−, and this difference significantly increased as conditioning progressed (CS+ responses during ACQ 1 v. ACQ 2: *p* < 0.001; ACQ 1 v ACQ 3: *p* < 0.001; ACQ 2 v. ACQ 3: *p* = 0.001) (effect of stimulus: *F*(1,194) = 975.02, *p* < 0.001, η^2^_p_ = 0.83; effect of trial: *F*(1.43,277.74) = 4.87, *p* < 0.01, η^2^_p_ = 0.02; Stimulus × Trial interaction: *F*(1.60,310.10) = 109.37, *p* < 0.001, η^2^_p_ = 0.36). Females also exhibited significantly greater overall US expectancy ratings than males (effect of sex: *F*(1,194) = 3.88, *p* = 0.05, η^2^_p_ = 0.02). No other main effects or interactions were significant (all *F* < 2.30, all *p* > 0.11).

### 3.3. Fear Generalization

#### 3.3.1. Fear-Potentiated Startle

Baseline startle responses to the first three NA trials on Day 2 decreased across trials (t1 v. t2: *p* = 0.045; t1 v. t3: *p* = 0.01) (effect of trial: *F*(2,400) = 5.11, *p* < 0.01, η^2^_p_ = 0.03). Similar to the analysis of Day 1 baseline startle responses, there was a significant effect of condition, *F*(2,200) = 3.22, *p* < 0.05, η^2^_p_ = 0.03; however, after applying the Bonferroni correction for post hoc comparisons, there were no significant differences observed. No other main effects or interactions were significant (all *F* < 1.97, all *p* > 0.14).

Analysis of fear-potentiated startle responses during generalization testing revealed a significant effect of stimulus, *F*(6.92,1384.26) = 15.15, *p* < 0.001, η^2^_p_ = 0.07. This effect revealed a typical generalization gradient, with the greatest fear-potentiated startle responses occurring following presentation of the CS+ (see Figure 5). Fear-potentiated startle responses decreased as the stimuli more closely resembled the CS−. No other main effects or interactions were significant (all *F* < 1.77, all *p* > 0.17). 

#### 3.3.2. Skin Conductance Response

Similar to the analysis of fear-potentiated startle, the analysis of SCRs during generalization testing revealed a significant effect of stimulus, *F*(7.24,1534.99) = 28.98, *p* < 0.001, η^2^_p_ = 0.12 (see Figure 5). This effect revealed a typical generalization gradient, with the greatest SCRs occurring following presentation of the CS+. SCRs decreased as the stimuli more closely resembled the CS−. No other main effects or interactions were significant (all *F* < 2.33, all *p* > 0.09). 

#### 3.3.3. US Expectancy Ratings

The analysis of US expectancy ratings during generalization testing revealed a significant effect of stimulus, *F*(2.76,574.10) = 120.33, *p* < 0.001, η^2^_p_ = 0.37 (see Figure 4). This effect revealed a typical generalization gradient, with the greatest US expectancy ratings occurring following presentation of the CS+. US expectancy ratings decreased as the stimuli more closely resembled the CS−. No other main effects or interactions were significant (all *F* < 1.22, all *p* > 0.24).

## 4. Discussion

The purpose of the present study was to examine the impact of acute stress administered 30 min prior to fear conditioning on the acquisition and generalization of fear. As expected, stress led to significant increases in HR, salivary cortisol levels, and subjective pain and stress ratings of the SECPT. To probe the involvement of corticosteroids in any stress-induced alterations of fear learning or fear generalization, we divided participants into cortisol “responders” and “non-responders” based on their salivary cortisol responses to the stressor. Upon examining the Day 1 data after this categorization, we found that the impact of stress on fear learning depended on the dependent measure that was used to quantify fear. Specifically, the analyses of fear-potentiated startle responses suggested that cortisol responders, but not non-responders, had impaired fear learning. In contrast, the analyses of skin conductance responses indicated that stress in general (i.e., both cortisol responders and non-responders) demonstrated impaired acquisition of fear. The analyses of Day 2 data indicated that pre-learning stress had no significant impact on fear generalization. These effects on fear learning were observed in the absence of any group differences for subjective US expectancy ratings. Overall, our findings support the hypothesis that when stress is temporally separated from conditioning, it impairs fear learning, potentially through corticosteroid-related mechanisms.

When we examined fear-potentiated startle responses across all three blocks of conditioning, cortisol responders exhibited impaired fear learning, as they failed to exhibit a greater response to the CS+ than to the CS−. Conversely, the analysis of SCRs suggested that stress, overall, impaired fear learning, as both cortisol responders and non-responders failed to differentiate between the CS+ and CS− during acquisition. Most previous work examining the impact of stress or stress hormones on fear conditioning has measured SCRs to quantify fear (e.g., [21,22,24,25,70,71,72]). Using fear-potentiated startle, in addition to SCRs, as a measure of fear in the present study provides a distinct advantage over using SCRs alone because, unlike SCRs, fear-potentiated startle is directly coupled with amygdala activity and provides a more direct measure of fear [73,74]. The observation that stress and, more specifically, cortisol responders exhibited reduced fear learning, as measured by fear-potentiated startle responses, is consistent with work from Antov et al. [22] and Merz et al. [21], who both reported that when pre-learning stress was temporally separated from fear conditioning, participants’ cortisol responses negatively correlated with their fear memory. The present findings thus support the hypothesis that when stress is administered 30 min before learning, it exerts deleterious effects on cognitive processes via a delayed (i.e., 20–30 min) increase in corticosteroid levels exerting a negative impact on synaptic plasticity in the amygdala and hippocampus.

The analysis of Day 2 data revealed that stress had no significant impact on the generalization of fear. This was unexpected, especially in light of the observed effects of stress on fear acquisition. We had hypothesized that stress would impair and/or slow fear learning and that this would result in a less-specific fear memory the next day. We did observe a relatively high non-response rate for our stress manipulation (i.e., the SECPT). Specifically, out of 93 stressed participants, only 27 (approximately 30%) demonstrated an increase of at least 1.5 nmol/l in salivary cortisol 25 min following stress onset. It is possible that, if we had obtained saliva samples longer after the onset of stress (e.g., 30–45 min post-stress), the number of responders would have increased. It is also possible that, in the present sample, the stress manipulation did not adequately activate the hypothalamus–pituitary–adrenal (HPA) axis to the degree that we have observed in our previous work. It is noteworthy to mention that examining the impact of stress on fear generalization is a relatively new area of research. Few studies have exposed participants to stress prior to fear conditioning and assessed its effects on fear generalization. Similar to our findings, Sep and colleagues [28] reported that stress administered 2 h before conditioning had no impact on fear generalization in a healthy participant sample. On the other hand, one preclinical study reported that stress administered 24 h prior to fear learning enhanced fear generalization in mice, and van Ast and colleagues [75] found that cortisol administration (note, however, that cortisol administration is not equivalent to stress) approximately 45 min before fear conditioning led to enhanced fear generalization in healthy participants, but only in females. Studying the impact of stress on fear generalization is applicable to our understanding of stress-related psychological disorders, such as PTSD. Investigators have speculated that PTSD may develop, at least in part, from an impaired ability to properly contextualize fear learning (e.g., [76]). In other words, PTSD patients fail to successfully differentiate between threatening stimuli (CS+) and stimuli that signal safety (CS− or GS). This results in a greater generalization of fear to non-threatening stimuli. Because of the importance of this research, additional work is needed to clarify the impact of stress on fear generalization in human participants.

Past research has shown that when stress is administered 30 or more minutes before learning, long-term memory is impaired. In theory, such stress impairs the function of cognitive brain regions, such as the hippocampus, via increased corticosteroid levels exerting inhibitory influences on synaptic plasticity. This would result in impaired pattern separation abilities, which have been proposed to underlie generalization phenomena and depend on hippocampal function [33,42,43]. For instance, Lissek and colleagues found that as the difference between generalization stimuli and the CS+ increased, so did activity in the hippocampus, as well as functional connectivity between the hippocampus and prefrontal cortex (PFC) [77]. Similar results have been reported by others [78,79,80]. Considering that Dunsmoor and colleagues observed a positive correlation between generalized fear and amygdala activity [81], the collective findings suggest that the hippocampus might facilitate stimulus discrimination by activating PFC mechanisms that send inhibitory output to amygdala fear centers [80]. Based on this reasoning, stress-induced impairments of hippocampal function should result in pattern separation deficits, leading to increased generalization of fear. Indeed, research has revealed that people with PTSD, a disorder linked to impaired hippocampal function, perform poorly on tasks involving pattern separation, exhibit an overgeneralization of fear, and display abnormal PFC function during generalization testing [31,33,82]. Individuals with hippocampal atrophy also exhibit impaired generalization [83,84], and lesioning the hippocampus [85] or cortical inputs to the hippocampus [86] in rats results in greater fear generalization. Thus, increased corticosteroid signaling as a result of stress that is temporally separated from learning could impair hippocampal function and pattern separation abilities, resulting in a weaker, non-specific fear memory, but this needs further evaluation in future work.

There are some limitations and caveats regarding the present study that warrant consideration. As noted above, we observed a relatively high non-response rate, in terms of salivary cortisol levels, for our stress manipulation (i.e., the SECPT). This could be the result of collecting too few saliva samples from participants and not quantifying their cortisol levels long enough after stress onset. In future work, it would be useful to generate a lengthier timeline of participants’ cortisol responses to the stressor. It is also possible that, in the present study, the SECPT did not adequately activate the HPA axis in a majority of participants. Thus, future studies may benefit from examining the impact of other stressors that are well known to generate robust cortisol responses, such as the Trier Social Stress Test [87], on fear learning and fear generalization. We also unexpectedly observed greater US expectancy ratings for the CS+, relative to the CS−, during the CS habituation phase of acquisition. It is possible that this effect occurred due to the CS+ always being presented first to participants. As participants were unsure what to expect at the beginning of the phase, they may have assigned a greater rating to the first stimulus observed. In future work, it may be helpful to modify the trial order to avoid such an effect. Finally, we observed several sex-dependent effects (e.g., subjective pain ratings, overall SCR magnitudes, Day 2 cortisol) in our study. We never observed significant interactions between sex, condition, and stimulus for fear acquisition or generalization, but this could be due to smaller sample sizes in our responder condition. It would be important for future work to examine more thoroughly the impact of sex on stress-induced alterations of fear learning and fear generalization, particularly because females are more likely than males to develop PTSD [88,89] and exhibit more intrusions of emotional stimuli than males [90,91,92,93].

## 5. Conclusions

In the present study, we have shown that acute stress administered 30 min before fear conditioning results in impaired fear learning but has no significant impact on fear generalization tested 24 h later. The effect of stress on fear learning was particularly evident in participants who exhibited a robust cortisol response to the stressor, which is consistent with theories regarding the time-dependent effects of pre-learning stress on cognition. Nevertheless, it is important to note that the effects of stress on fear conditioning were not completely consistent across our dependent measures (i.e., fear-potentiated startle and SCRs), and, following categorization of stress participants based on their cortisol response to the stressor, the sample size of cortisol responders was only 27. Therefore, our findings should be considered preliminary and warrant further investigation to aid our understanding of how pre-learning stress impacts fear learning and fear generalization.

## Figures and Tables

**Figure 1 biology-12-00775-f001:**
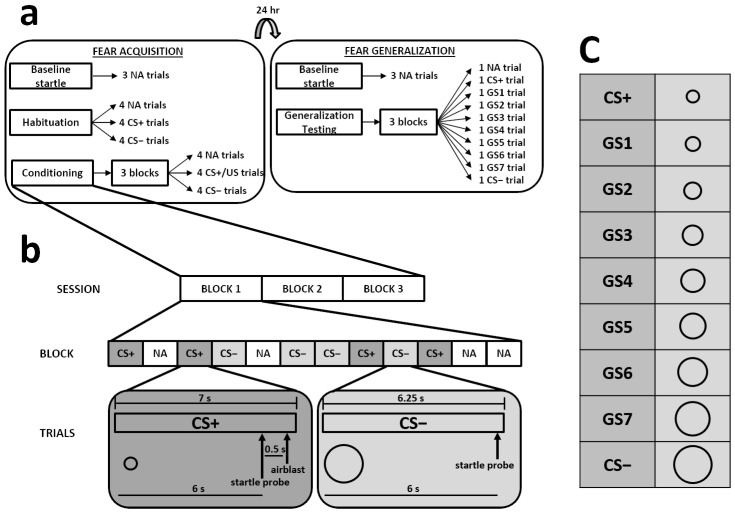
The top part of the figure illustrates the fear-potentiated startle paradigm (**a**). The acquisition and generalization sessions were separated by 24 h, and each began with 3 noise-alone (NA) trials. The acquisition session included a habituation phase (no stimulus was followed by the US) and a conditioning phase (the CS+ was always reinforced with the US). The generalization session included 3 blocks of NA, CS+, GS, and CS− trials; none of the trials involved presentation of the US. The lower part of the figure depicts the composition of the trial blocks from the conditioning phase of acquisition (**b**). In this example, the small circle is shown as the CS+, and the large circle is shown as the CS−. Within each block, participants were exposed to 12 trials, 4 of each trial type (i.e., CS+, CS−, NA). The timelines for CS+ and CS− exposure, relative to the startle probe and US, are illustrated below these trial types. The right part of the figure depicts the generalization stimuli (GS) that were used during the generalization session on Day 2 and their size compared to the CS+ and CS− (**c**).

**Figure 2 biology-12-00775-f002:**
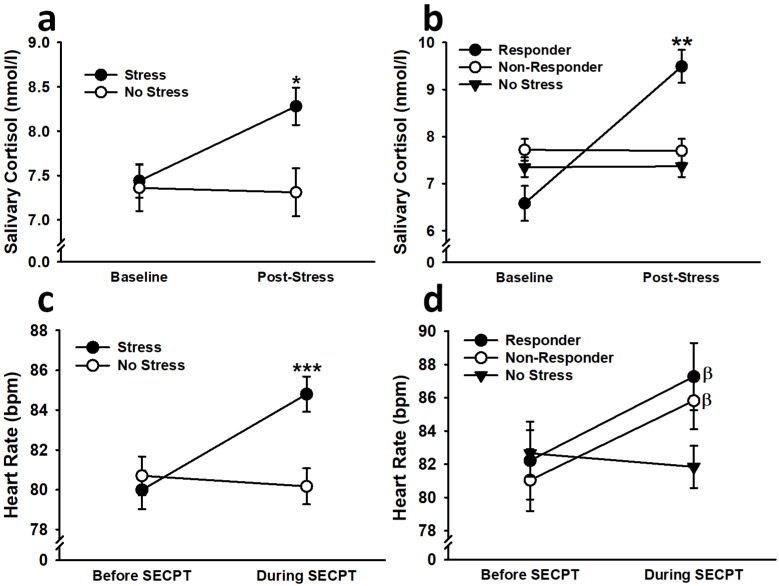
Stressed participants exhibited greater salivary cortisol levels than non-stressed participants following exposure to the SECPT (**a**). After stressed participants were divided into “responders” and “non-responders”, this increase in salivary cortisol levels, relative to non-stressed participants, was observed in responders only (**b**). Stressed participants also exhibited greater HR than non-stressed participants following the SECPT (**c**). After dividing stressed participants into cortisol responders and non-responders, the analysis showed that both responders and non-responders exhibited increases in HR following the SECPT, while non-stressed participants did not (**d**). Data are presented as means ± SEM. * *p* < 0.01 relative to no stress and before SECPT; ** *p* < 0.01 relative to no stress, non-responder, and before SECPT; *** *p* < 0.05 relative to no stress and before SECPT; β *p* < 0.05 relative to before SECPT.

**Figure 3 biology-12-00775-f003:**
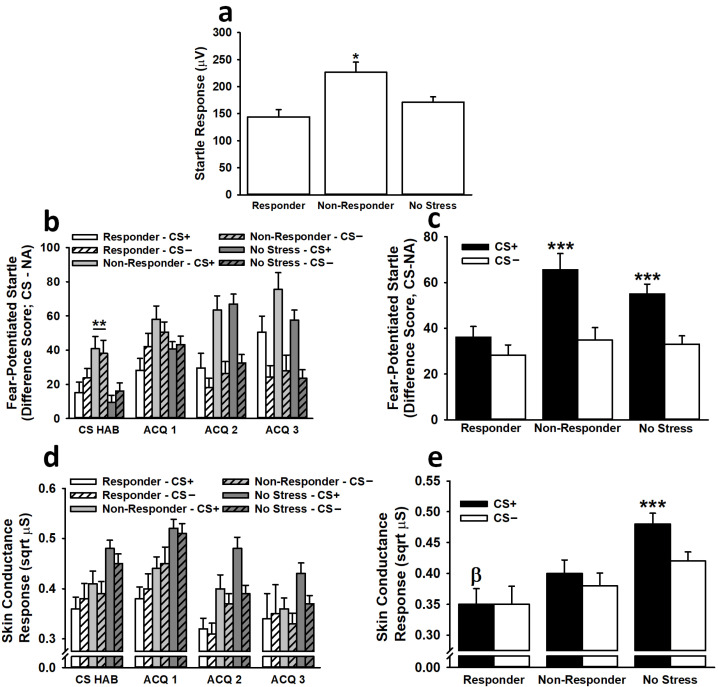
During fear acquisition on Day 1, non-responders displayed greater baseline startle responses than responders and non-stressed participants (**a**). Non-responders also exhibited greater fear-potentiated startle responses, overall, than non-stressed participants during the CS habituation phase (**b**). In general, during the 3 conditioning blocks (ACQ 1–ACQ 3), participants exhibited greater fear-potentiated startle and skin conductance responses to the CS+ than to the CS−; however, these effects depended on the condition. Across all 3 conditioning blocks, non-responders and non-stressed participants demonstrated greater fear-potentiated startle responses to the CS+ than to the CS−, while responders did not (**c**). Similarly, across all 3 conditioning blocks, non-stressed participants displayed greater skin conductance responses to the CS+ than to the CS−, while responders and non-responders did not (**d**,**e**). Data are presented as means ± SEM. * *p* < 0.05 relative to responders and no stress; ** *p* < 0.001 relative to no stress; *** *p* < 0.001 relative to CS−; β *p* = 0.016 relative to no stress.

**Figure 4 biology-12-00775-f004:**
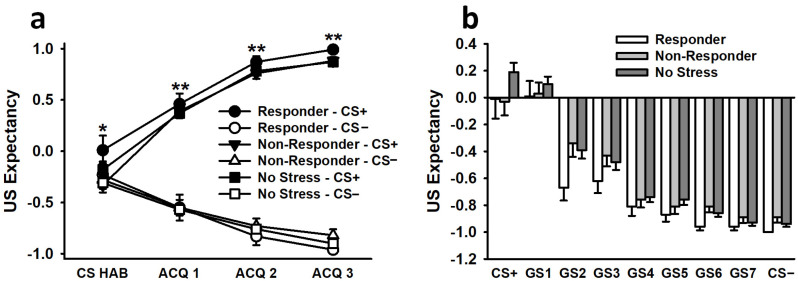
Participants exhibited greater US expectancy ratings for the CS+ than for the CS− during the CS habituation phase and during all 3 blocks of conditioning on Day 1 (**a**). During the generalization phase on Day 2, US participants displayed expectancy ratings that reflected a typical generalization gradient (**b**). The greatest ratings were observed following exposure to the CS+, and these ratings lessened as the stimuli more closely approximated the CS−. Data are presented as means ± SEM. * *p* < 0.05 CS+ relative to CS−; ** *p* < 0.001 CS+ relative to CS−.

**Figure 5 biology-12-00775-f005:**
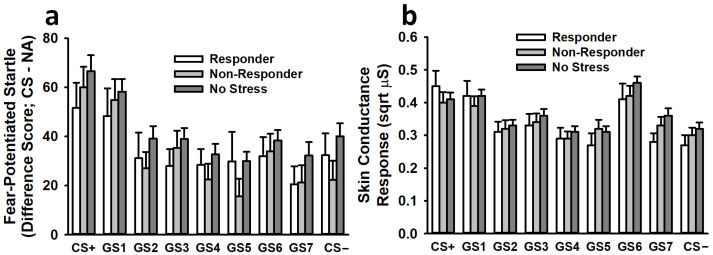
During the generalization phase on Day 2, participants exhibited fear-potentiated startle (**a**) and skin conductance (**b**) responses that reflected typical generalization gradients. The greatest responses were observed following exposure to the CS+, and these responses weakened as the stimuli more closely approximated the CS−. Pre-learning stress had no significant impact on these generalization gradients. Data are presented as means ± SEM.

## Data Availability

Data reported in this manuscript are a subset of the data found in Collection 3106 of the NIMH Data Archive.

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
