# Peer review of "Pre-Learning Stress That Is Temporally Removed from Acquisition Impairs Fear Learning"

_biology, 2023, doi:10.3390/biology12060775_

Round 1

Reviewer 1 Report

The current study sought to examine the relationship between stress, cortisol levels, and fear acquisition/generalisation. Although overall I enjoyed reading the manuscript (the presentation and writing style are great) – a number of critical issues that need to be addressed, especially with the analysis and interpretation of the results:

Is there a citation for the SCR scoring method used? This is an unusual baseline correction that I have not seen before. What are the logistical issues with measuring both FPS and EDA concurrently (i.e., presentation of startle probe will affect SCRs)? How were these issues resolved?

In the methods it states “Participants exhibiting a cortisol increase of at least 1.5 nmol/l following the SECPT were considered Responders (N = 27; 13 males, 14 females); all other stressed par- ticipants were considered Non-Responders (N = 66; 24 males, 42 females)” – I may have missed it but why are the numbers for responders + non-responders so low compared to the group sizes for the stress condition (N=113)? Also, this is a very high rate of non-responding. What is the explanation for this? My first thought was the relatively early measurement of peak cortisol (25-min post stress), when oftentimes peak cortisol can be seen in many individuals at least 30 minutes post stress, particularly with physical stressors.

The posthoc analyses used to interpret trial * condition * stimulus effects for acquisition (for SCR and FPS) are fundamentally flawed. The authors report non-significant interaction effects followed by Bonferroni-corrected follow up tests that show that significant differences are observed in one group (between CS+ and CS-) but not others. Regardless of whether Bonferroni corrections are applied or not, this follow-up test does not show any difference at all between the groups across trials, as it is only comparing the CS+ and CS- within each group. For example, it is possible within this data that the difference between responder differential FPS responding on Block 3 was not significant at p = .051 but the non-responders on block 2 & 3 were significant at p = .049 (see Nieuwenhuis et al 2011 Nature Neuroscience for a discussion on this issue). Further to this, the p values and effect size differences for these follow up tests are not reported so this assessment by a reader cannot be made.

The same error is made in the generalisation section: “In fact, non-responders and non-stressed participants demonstrated significantly greater fear-potentiated startle responses to the CS+ than to all other stimuli (all p < 0.05), except for GS1, while respond-ers did not exhibit significantly greater startle responses to the CS+ than to any other stim-ulus presented during generalization (all p > 0.09).” These p values seem to instead suggest that there was no difference between the groups, and this is supported by the actual tests reported earlier in this section.

In light of both of these issues, the discussion and abstract and all related interpretations of these interactions should be completely revised as they are (most likely) inaccurate. My reading of the data is that the authors find acquisition effects (of cortisol responsiveness, with no associated trial effects) but no effects of generalisation. I understand that this is a significant revision to the interpretation of the results, but I don’t think that there is anything wrong given the study design with finding a non-significant effect.

All of the acquisition analyses include the first block, which has no difference. I am assuming no difference here is because the EDA and FPS measurements are taken BEFORE the first US is presented - this means that no acquisition learning could have occurred on the first trial. I would suggest re-running these analyses with the first trial excluded.

In the SCR section: “Alt-hough the Condition x CS x Trial interaction was not significant, F(3.76,358.62) = 0.20, p > 0.93, η2p = 0.00, we performed Bonferroni-corrected comparisons between fear-potentiated startle responses to the CS+ and CS- for each condition during each trial block”. I assume that FPS here should be SCR?

Could the US expectancy effect during habituation be due to trial order effects? (i.e., CS+ always/usually presented first)?

The authors state that the results of stress induction were not significant but (unless I missed them) these results are nowhere reported. The addition of a supplementary material with these results would be helpful.

Author Response

Please find our responses in the attached file.

Reviewer 2 Report

The present study by Zoladz et al. investigated how a brief stressor (Socially evaluated cold pressor test, SECPT) prior to an associative fear learning task affects the acquisition and generalization of fear, thereby aiming to provide further insight into the etiology of stress-related psychological disorders. The authors found that a robust cortisol response to the stressor (‘responders’) was associated with slower fear learning and increased fear generalization. I recommend this article for publication in Biology. However, the manuscript should be revised with a focus on following issues:

(1)   The authors state that their data supports the hypothesis that when stress is temporally separated from learning, the developing fear memory is weakened through corticosteroid-related mechanisms. Could the authors specify why they decided to start the fear acquisition session 30 minutes after the SECPT stressor? Is there any data available for other time points?        

(2)   Why do the authors conclude that stress exerts impairing effects on fear learning via a ‘delayed increase in corticosterone levels’ (l. 535)? Did they perform a time course study? Did the studies cited use the same temporal window?        

(3)   There seem to be quite some measurements which show sex-dependent effects (e.g., Painfulness, SCR, salivary cortisol levels after generalization testing, US expectancy). Could the results observed be driven by sex?     

(4)   Did the authors find differences in the data of participants that removed their hand from the water before the 3 minutes had passed (n = 25) or were these excluded?        

(5)   How do the authors explain the higher stressfulness scoring of stressed ‘non-responders’? Have the authors performed an analysis where they used the individual’s stress rating as categorization into responders and non-responders?

(6)   From the data shown, ‘increased fear generalization’ is a pretty strong statement. Consider re-phrasing.    

(7)   Please include the references of other published manuscripts that utilized the same dataset. 

(8)   Typo l. 535: ‘increase’ instead of ‘increased

-

Author Response

(The authors gave the same response as above.)

Round 2

Reviewer 1 Report

Good revision - no further comments